# Differential Expression of miRNAs Contributes to Tumor Aggressiveness and Racial Disparity in African American Men with Prostate Cancer

**DOI:** 10.3390/cancers15082331

**Published:** 2023-04-17

**Authors:** Richard Ottman, Kavya Ganapathy, Hui-Yi Lin, Carlos Diaz Osterman, Julie Dutil, Jaime Matta, Gilberto Ruiz-Deya, Liang Wang, Kosj Yamoah, Anders Berglund, Ratna Chakrabarti, Jong Y. Park

**Affiliations:** 1Burnett School of Biomedical Sciences, University of Central Florida, Orlando, FL 32816, USA; rjottman@gmail.com (R.O.); kaganapathy@knights.ucf.edu (K.G.); 2Biostatistics Program, School of Public Health, Louisiana State University Health Sciences Center, New Orleans, LA 70112, USA; hlin1@lsuhsc.edu; 3Department of Basic Sciences, Ponce Research Institute, School of Medicine, Ponce Health Sciences University, Ponce, PR 00716, USA; cjdiaz@psm.edu (C.D.O.); jdutil@psm.edu (J.D.); jmatta@psm.edu (J.M.); gruiz@psm.edu (G.R.-D.); 4Department of Tumor Biology, H. Lee Moffitt Cancer Center, Tampa, FL 33612, USA; liang.wang@moffitt.org; 5Department of Radiation Oncology, H. Lee Moffitt Cancer Center, Tampa, FL 33612, USA; kosj.yamoah@moffitt.org; 6Department of Biostatistics and Bioinformatics, H. Lee Moffitt Cancer Center, Tampa, FL 33612, USA; anders.berglund@moffitt.org; 7Department of Cancer Epidemiology, H. Lee Moffitt Cancer Center, Tampa, FL 33612, USA

**Keywords:** prostate cancer, miRNA, health disparity, aggressiveness

## Abstract

**Simple Summary:**

Prostate cancer (PCa) is the leading cancer in incidence and second leading cause of cancer mortality in US men. Recent data showed a 3% increase in PCa incidence rate each year from 2014 through 2019. African American (AA) men have 1.6-fold higher incidence and 2.2-fold higher PC mortality rates than European American (EA) men. Growing evidence shows that miRNAs are closely associated with aggressiveness and racial disparity in prostate cancer and might facilitate the prediction of prognosis and a treatment plan. In this study, we identified differentially expressed miRNAs, which are significantly correlated with the aggressiveness and health disparity of prostate cancer. These findings may assist personalized medicine, suggesting miRNAs as promising biomarkers for prostate cancer, especially in African American men.

**Abstract:**

Prostate cancer is the leading cancer in incidence and second leading cause of cancer mortality in US men. African American men have significantly higher incidence and mortality rates from prostate cancer than European American men. Previous studies reported that the disparity in prostate cancer survival or mortality can be explained by different biological backgrounds. microRNAs (miRNAs) regulate gene expression of their cognate mRNAs in many cancers. Therefore, miRNAs may be a potentially promising diagnostic tool. The role of miRNAs in prostate cancer aggressiveness and racial disparity has not been fully established. The goal of this study is to identify miRNAs associated with aggressiveness and racial disparity in prostate cancer. Here we report miRNAs that are associated with tumor status and aggressiveness in prostate cancer using a profiling approach. Further, downregulated miRNAs in African American tissues were confirmed by qRT-PCR. These miRNAs have also been shown to negatively regulate the expression of the androgen receptor in prostate cancer cells. This report provides a novel insight into understanding tumor aggressiveness and racial disparities of prostate cancer.

## 1. Introduction

Prostate cancer (PCa) is the leading cancer in incidence and second leading cause of cancer mortality in US men [1]. In 2023, new prostate cancer cases and deaths expected in the US are 288,000 and 34,700, respectively. In addition, recent data showed a 3% increase in PCa incidence rate each year from 2014 through 2019 [2]. African American (AA) men have 1.6-fold higher incidence and 2.2-fold higher PC mortality rates than European American (EA) men [3]. Previous studies have reported that PCa survival or mortality disparities cannot be fully explained by different socioeconomic status [4,5]. These results suggest that biological background accounts for a significant portion of PCa disparity with regard to mortality, incidence, and progression in AA men compared to EA men. However, further investigation is required to uncover the mechanisms underlying abnormal gene regulation and racial disparity. Also, there is an unmet need to develop prognostic biomarkers that enable the reduction of AA PCa racial disparities.

MiRNAs are endogenous, short (19–24 nucleotides) non-protein-coding RNAs that regulate gene expression at the posttranscriptional level via binding to 3′-untranslated regions of protein-coding transcripts [6]. MiRNAs are pleiotropic in terms of functions; they regulate the expression of a broad range of genes involved in cancer [7]. MicroRNAs (miRNAs) are involved in gene silencing through inhibition of translation and destabilization of mRNAs, and thereby regulation of a variety of signaling pathways [8]. Dysregulated expression of miRNAs contributes to the abnormal expression of mRNAs which mediates phenotypic changes in various cancers [9]. Recently, several studies were reported on the role of miRNAs in risk, progression, and prognosis of prostate cancer. miR-5100, miR-199b-3p, miR-26b-5p, and miR-98-5p were associated with the risk of prostate cancer [10,11,12]. In addition, miR-26b-5p, miR-4732-3p, miR-181A, miR-205, miR-3195, and miR-4417 were suggested as potential biomarkers for differentiating advanced cases from an early stage of prostate cancer [11,13,14]. miRNA-532-5p, miR-17-5p, and miR-199b-3p were proposed as biomarkers for prognosis [12,15,16]. However, the role of miRNA-mediated gene expression regulation in the biological contribution to the observed racial disparities in prostate cancer has not been established. Thus, the goal of this study is to identify miRNAs involved in the racial disparities of PCa. Additionally, these miRNAs may be a risk factor for poor prognosis among AA patients. Thus, regulation of these miRNAs may offer a preventative and therapeutic approach for men at risk of PCa in the AA population.

## 2. Materials and Methods

### 2.1. Patient Selection and Procurement of Human Prostate Tissues

Prostate tissues obtained by radical prostatectomies were procured in the Cooperative Human Tissue Network (Southern division) at the University of Alabama at Birmingham (UAB) in accordance with an approved IRB protocol. Formalin-fixed and paraffin-embedded (FFPE) tissues from 25 PCa patients, including 10 AA patients, were evaluated by a pathologist, and tumor and adjacent non-involved areas were macro-dissected for RNA extraction followed by qRT-PCR for the expression of candidate miRNAs in patient tissues, as described previously [17,18]. Cases were selected based on the Cancer of the Prostate Risk Assessment (CAPRA)-S score, a prognostic tool for predicting a patient’s risk for biochemical failure following radical prostatectomy [19].

### 2.2. RNA Extraction and cDNA Synthesis

RNA extraction from FFPE tissue sections was conducted using the RecoverAll kit (Life Technologies, Carlsbad, CA, USA). Before RNA was extracted, the tissue sections were evaluated for the presence of tumor lesions. Adjacent non-involved tissue blocks were used for the control group. The non-involved tissue sections were evaluated for the presence of tumor lesions by a pathologist. If tumor cells were observed in more than 10% of the section, additional sections were evaluated in a similar manner. Total RNA was isolated from 20 µm thick sections from tissue blocks and used for subsequent cDNA synthesis using the QuantiMir RT kit (System Biosciences, Palo Alto, CA, USA). Poly-A tail synthesis was first conducted using PolyA polymerase, and oligo dT anchor was annealed to the RNAs. RNA samples were next used for reverse transcription and quantitative RT-PCR (qRT-PCR).

### 2.3. Quantitative Real-Time PCR

The expression of mature miRNAs from FFPE tissues was determined by using the miRNome miRNA Profiling Kit (System Biosciences). The kit provides specific primers for 1,113 mature miRNAs and includes primers for 3 internal control RNAs (U6 snRNA, RNU43 snoRNA, RNU1A snRNA). MiRNA IDs listed in the text are based on Sanger miRBase identifiers. Primers were designed to maintain uniform amplification efficiencies. qRT-PCR reaction mixtures were prepared using 2X Maxima SYBR Green/ROX qPCR Master Mix (Thermo Fisher Scientific, Waltham, MA, USA). For profiling and validation, qRT-PCR was conducted using the 7900HT thermal cycler (Applied Biosystems Inc., Foster City, CA, USA). The data were initially analyzed using the SDS v2.3 software (ABI). DNA concentrations were reported through SYBR Green fluorescence and normalized to that of the passive reference dye, ROX. Ct values calculated by the SDS 2.3 software were transferred to the miRNome analysis software (SBI) to derive ∆∆Ct values. The miRNome analysis software calculates the ∆Ct values based on the mean of the reference genes. The individual ∆Ct values are then compared across samples to generate the ∆∆Ct values for each miRNA. miRNAs that showed significant changes in expression were then subject to further analysis. The statistical analysis of the qRT-PCR data is described below. We used macro-dissected prostate tumor tissues and corresponding adjacent uninvolved areas to monitor the expression of mature miRNAs. Patients were selected based on specific criteria including no prior treatments, Gleason scores, pre-surgical prostate-specific antigen (PSA), local invasion, and CAPRA-S score [19] stratified into low, medium, and high risk of biochemical recurrence (Table 1). We used a profiling approach with miRNome miRNA profiling kit (System Biosciences) to identify miRNA expression patterns for each patient’s tumor and associated adjacent uninvolved prostate tissue.

The analysis of miRNA expressions stratified by CAPRA-S score identified patterns consistent with our hypothesis and previous reports [10,11].

### 2.4. Data Analysis

Normalization of qRT-PCR expression values was further refined using the qBasePlus software 2.0 (Biogazelle: Zwijnaarde, Oost-Vlaanderen, Belgium). Using the Genorm functionality included with the qBasePlus software, 7 additional stably expressed miRNAs were identified. The ∆Ct value for each miRNA was then re-calculated utilizing the 7 additional miRNAs plus the 3 original controls. Following normalization, the expression of the reference miRNAs was re-evaluated to ensure their stability across samples was maintained. For the fold change, values were calculated next to determine expression differences between paired adjacent uninvolved and tumor tissues. After normalization, the software assigned relative expression values where the mean expression of the reference genes is determined to be a value of 1. The expression of each miRNA is then assigned a relative expression value with respect to the geometric mean of the controls. The geometric means were compared using the formula: ΔCt control = 2^−(GMc-GMr)^
(1)
where GMc is the geo-mean of the control sample and GMr is the geo-mean of the reference sample. The fold change or ∆∆Ct, for each miRNA was then calculated using the formula:∆∆Ct = 2^−(CtR-CtC)^ × (∆Ct control)(2)
where CtR = Reference sample miRNA Ct value, and CtC = Control sample miRNA Ct value.

Additional analysis was conducted using Cluster 3.0 software. For cluster analysis, log2 transformed normalized Ct values for each miRNA in each tumor, and uninvolved samples were used in an expression matrix where each miRNA is presented in rows and samples are presented in columns. For hierarchical clustering, we used a gene-centric (miRNA) approach to analyze and display the expression pattern upon centering; which, shows the relative up–down expression pattern for a particular miRNA across the samples based on its median expression value in shades of red (up) and green (down). The results of cluster analysis are displayed as heat maps generated by Java TreeView software. Heatmaps of differentially expressed miRNAs are created for viewing similar miRNA groups in a dataset using Pearson correlation with Average Linkage.

### 2.5. Statistical Analysis

Statistical analysis of the expression data was performed using log2 transformed Ct values using GraphPad Prizm. The *p*-values were calculated by the Mann–Whitney U test to estimate the statistical significance between the two groups.

## 3. Results

### 3.1. miRNA Expressions Deregulated in Human Prostate Tumors

Normalized relative expression values were used for Cluster analysis. Hierarchical clustering of the normalized and log2-transformed expression data showed four distinct clusters of miRNAs (Figure 1).

Clusters 1 and 3 identified miRNAs distinctly expressed between malignant and uninvolved tissues (Figure 2A). The Log2-transformed relative expression values were extracted from each cluster and the average expression of each miRNA was calculated for both uninvolved tissue and tumor tissue samples. The average relative expression for individual miRNAs (unidentified) in both uninvolved and tumor groups is presented in Figure 2B,C for clusters 1 and 3, respectively. The jittered strip chart displays the distribution of average miRNA expression values for both groups including the average of all miRNAs in the cluster (black bars) ±1 SD. In cluster 1 (Figure 2B), we identified 24 miRNAs (average values) that displayed downregulation in tumor tissues compared to uninvolved tissues. Examination of this cluster distinguished six miRNAs (miR-143. -133a, -133b, -204, -221, -222) with on average greater than 2-fold downregulation (*p* = 0.0002, 1.68 × 10^−5^, 0.005, 0.0009, 1.17 × 10^−5^, 0.0005, respectively) in expression in malignant tissues. Alternatively, the trend of average miRNA expression presented in cluster 3 identified 17 miRNAs with increased expression in tumor tissues compared to uninvolved tissues (Figure 2C). Cluster 3 also contained a subset of six miRNAs that displayed on average a change in expression greater than 2-fold. These six miRNAs (miR-375, -183, -93, -96, -127-5p, and -380) expressed higher levels in malignant tissues compared to uninvolved tissue samples (*p* = 0.06, 0.0003, 0.0002, 0.032, 0.012, 0.013, respectively).

### 3.2. Deregulation of miRNA Expressions Associated with Aggressiveness of Prostate Tumors

Initial analysis of clusters 2 and 4, identified from miRNA array expression analysis (Figure 1), did not show the visibly unique expression patterns between tumor and uninvolved prostate tissues, as identified in clusters 1 and 3. The cluster 2 heat map (Figure 1) depicts similar levels of expressions for all miRNAs (miR-103, -107, -29a/b/c, -199a/b-3p, and let-7a/b/d/e/g/i) within individual patient tissue; while, miRNAs display heterogeneous expression across samples. Further interrogation of the cluster revealed a correlation of miRNA expression profiles with the patient’s CAPRA-S score (Figure 3C). Fifty percent (9/18) of tumor tissues from low- and medium-risk patients (CAPRA-S score 0–5) expressed cluster-specific miRNA averages below the average expression observed in all benign samples. However, 83% (5/6) of tumor tissues from high-risk patients (CAPRA-S score ≥6) expressed cluster-specific miRNA averages below the average expression observed in all benign samples (Figure 3A,C). When the cluster was dissected into individual miRNAs, there was a 2-fold change in expression (±0.2) for nine of 13 miRNAs in cluster 2, while there was no significant reduction observed in low- and medium-risk patient tumors. In cluster 4, the trend of increasing miRNA expression correlates positively with increasing risk of disease recurrence as predicted by CAPRA-S score groups: low risk (CAPRA-S: 0–2), medium risk (CAPRA-S: 3–5), and high risk (CAPRA-S: ≥6) (Figure 3B,D). Based on the fold change in the expression of miRNAs stratified by CAPRA-S risk groups, and have identified the top 60 miRNAs with >1.5-fold changes in expression in high-risk groups (30 upregulated and 30 downregulated).

### 3.3. Differential Expression of miRNAs in Prostate Tumors from African American and European American Patients

Next, we sought to identify miRNA expression profiles differentially expressed in malignant prostate tissues from AA men, specifically. From the data generated in our miRNA profiling study, we compared the relative expression of miRNAs in prostate tumor tissues from AA and European American (EA) men. Our analysis identified miRNAs that, on average, exhibited a >2-fold difference in expression (increased and decreased) in AA compared to EA men. Expressions of some miRNAs, miR-541, -34c-5p, -135b, -299-3p, -491-5p, and -30e, were reduced in the tumors of AA men compared to EA men. These miRNAs also have been shown to negatively regulate the expression of the androgen receptor in PCa [20]. To better understand how these miRNAs are regulated, we examined the fold change in expression of these miRNAs in patient-specific tumor tissues compared with matched benign prostate tissue. The patients were grouped by race and subdivided by CAPRA-S score (0–3 or ≥4). This analysis highlighted the consistent pattern of downregulation of these six miRNAs in AA men. In comparison, EA patients displayed a much broader distribution in expression. No significant difference in expression was noted in samples with a CAPRA-S score lower than or equal to 3 (Figure 4A); while, a significant difference in expression of five miRNAs at a 5% level and one at a 10% level was noted in AA tumors with a CAPRA-S score ≥4 compared to EA tumors (Figure 4B).

## 4. Discussion

African American (AA) men are at a 2.2-fold increased risk of prostate-cancer-specific (PCa) mortality compared with European American (EA) men. However, the relationship between this observation and miRNAs, and how this relationship explains PCa racial disparities, is not well established. This study represents an ongoing effort to investigate miRNAs in AA men with PCa. We observed that several differentially expressed miRNAs were found in tumor tissue from AA PCa patients. Our findings may help us to understand potential mechanisms for tumor aggressiveness and racial disparity. Once we identify these unique miRNA profiles at diagnosis, this information may provide biomarkers to determine treatment strategies for men with aggressive PCa. In this study, we identified miRNAs associated with PCa, aggressiveness, and potential health disparities among AA men.

### 4.1. miRNAs Associated with Tumor Status

Several miRNAs showed at least a 2-fold difference in expression between the tumor and adjacent uninvolved tissues. These miRNAs are miR-375, -183, -93, -96, -127-5p, -380, -143, -133a, -133b, -204, -221, and -222. Among these miRNAs, several miRNAs have been extensively investigated in previous studies.

For example, miR-375 was identified as a biomarker with a high-level sensitivity and specificity in PCa detection [21]. Several studies have observed that miR-375 is upregulated in primary tumor tissues and serum [22,23]. Further, Schaefer et al. reported the combination of six miRNAs including that miR-375 was used; the AUC was significant (0.88) in discriminating normal and tumor tissue [24]. miR-375 is also associated with clinical variables. Including, a high Gleason score, lymph-node-positive status, biochemical recurrence, and metastasis [25,26,27]. Cheng et al. [28] observed upregulation of miR-375 in serum samples from patients with metastatic, castration-resistant PCa. These results were found in the screening cohort; the serum level of miR-375 was significantly increased in PCa cases as compared with the testing cohort (AUC = 0.77) and a validation cohort. Haldrup et al. [29] confirmed dysregulation of miR-375 using genome-wide miRNA profiling of serum samples and was able to identify 84% of all PCa patients. However, these findings of miR-375 in tumor tissue are not always consistent. Kachakova et al. reported that miR-375 was significantly downregulated in 83.5% of PCa patients compared to benign prostatic hyperplasia (BPH) controls [30]. Although functional studies might define certain miRNAs as onco-miRNAs or tumor-suppressor miRNAs, their expression in prostate tumors might not correlate with these classifications. For instance, normally, miR-375 is upregulated as an onco-miRNA in PCa tumor tissue relative to normal tissue. However, forced expression of miR-375 decreased proliferation and invasion of androgen-independent PC-3 cells [31]. 

The overexpression of miR-183 in PCa tissues was reported in previous studies [32,33,34]. Larne et al. proposed a formula that can predict aggressive progression characteristics. This formula, consisting of four miRNAs including miR-183, is associated with tumor grades, PSA levels, metastasis, and survival. More importantly, this signature distinguishes aggressive tumors from non-aggressive PCa with an Area under the ROC Curve (AUC) of 0.90 [35]. Martens-Uzunova et al. found that levels of several miRNAs, including miR-183, significantly differ in lymphocytes and could be used in the evaluation of the progression of PCa [36]. Recently, miR-183 was suggested as one of the oncogenic clusters for PCa after a series of analyses using various data including TCGA [37].

miR-96 was reported as one of the overexpressed miRNAs in malignant prostate tissue compared with normal adjacent prostate tissue [32,33,34,38,39,40]. Schaefer et al. reported that miRs-96 showed a significant correlation with the Gleason score. Furthermore, the combination of six miRNAs including miR-96 provided a significant AUC (0.88) in discriminating normal and tumor tissue [24]. Larne et al. developed a formula that can predict poor outcomes, such as grades, PSA level, metastasis, and survival. More importantly, this formula distinguishes aggressive tumors from non-aggressive PCa with an AUC of 0.90 [35]. Martens-Uzunova et al. found miRNA-96 expression was significantly different in lymphocytes from progressed PCa [36]. Recently, miR-96 was suggested as one of the onco-miRNAs after a series of bioinformatic analyses using various data including TCGA [37].

### 4.2. miRNAs Associated with Prognosis

Deregulated miRNA expression has been associated with tumor progression in PCa [41]. Previous studies reported differentially expressed miRNAs associated with PCa progression [21,42,43,44]. Although those results are not consistent, several miRNAs, such as miR-1, -145, -205, -221, and -375, were suggested as good candidates for the prognosis of prostate cancer [42,44,45]. We identified several miRNAs, miR-1, -127-5p, -139-5p, -145, 296-5p, -302a, -330-5p, -365, -495, -509-3-5p, -511, and -518d-5p, as potential biomarkers for prognosis in this study. Among these miRNAs, some miRNAs were evaluated in previous studies. 

miR-1 is known as an oncomiRNA, and is involved in bone metastasis by activating the epidermal growth factor (EGFR) [46]. Previous studies reported that the downregulation of miR-1 in PCa tissues is linked to PCa progression, castration-resistant disease, and metastasis [36,47]. Furthermore, downregulations of miR-1 contribute to the proliferation, migration, and invasion of PCa cells. Therefore, miR-1 was suggested as a candidate prognostic biomarker for PCa in previous studies [22,46,48,49,50,51,52].

miR-139-5p downregulation in prostate tumor tissue has been previously reported [36,53]. Prior findings indicate that miR-139-5p inhibits the proliferation of PCa cells by interfering with the cell cycle [54], functioning as a tumor suppressor in PCa through regulation of SOX5 [55].

miR-145 binds to the 3′UTR of *MYO6* and is regulated inversely, resulting in a decrease in myosin VI; which, is involved in cancer-related cell migration and β-actin in the LNCaP PCa cell line [23]. Ectopic expression of miR-145 in LNCaP cells significantly reduced the proliferation [21,56,57,58]. miR-145 expressed in endothelial cells of blood vessels but not stromal cells [21]. Several studies reported downregulation of miR-145 in prostate tumor samples [23,24,57,58,59,60,61] and metastatic lymph nodes [36]; especially miR-145, which showed significant downregulation in aggressive PCa [58]. The reduction of miR-145 expression was also correlated with clinical variables, such as the Gleason score, clinical stage, tumor size, and PSA level. Further, miR-145 expression was correlated with risk for biochemical recurrence and shorter disease-free survival [35,50,62,63]. Based on these studies, miR-145 was suggested as a biomarker with a high-level sensitivity and specificity in PCa detection [21].

### 4.3. miRNA Associated with Racial Disparity

We are interested in investigating the role of miRNAs associated with health disparities in PCa. We used prostate tissue samples obtained from AA and EA patient cohorts. We used microarray analysis and qRT-PCR techniques to confirm our results.

The relative expression of miR-34c was significantly lower in the tumor tissues compared to the benign prostatic hyperplasia (BPH) tissues, and inversely correlated with a high Gleason score [39], PSA level, metastatic status, survival [34,64,65], clinical stage, and status of TMPRSS2-ERG [66]. Low expression of miR-34c was suggested to occur due to DNA methylation, loss of heterozygosity in the 11q23 region, or p53 mutation. Hagman et al. demonstrated that the function of miR-34c in PCa is mediated by targeting MET [67]. 

Previous studies have shown the tumor suppressor role of miR-30 in various cancers, with this miR-30e being extensively studied and well-characterized [17]. The relative expression of miR-30e was significantly lower in the tumor tissues compared to the normal tissues [23,33]. Recently, a multidimensional function of miR-30e through the regulation of genes involved in various signaling pathways was reported. Ganapathy et al. observed low expression miR-30e in prostate tumors and experimental upregulation led to cell cycle arrest, apoptosis, drug sensitivity of PCa cells, and reduced tumor progression [17]. 

miR-299-3p, another androgen receptor (AR) targeting miRNA, which showed downregulation in PCa from AA patients, has also been shown to function as a tumor suppressor in a number of cancers including colon cancer and hepatocellular carcinoma [68,69]. Recently, Ganapathy et al. showed loss of expression of miR-299 in prostate tumors, and restored expression of this miRNA improved drug sensitivity and exhibited a tumor suppressor function that is mediated through targeting AR and vascular endothelial growth factor (VEGF)A [18].

miR-135b has been reported to be downregulated in prostate cancer and to play a role in the progression of PCa. Tong et al. developed the 48-miRNA signature, including miR-135b, that predicted biochemical recurrence after prostatectomy [57]. Previous studies reported that downregulation of miR-135b was associated with the status of tumor and tumor metastasis [70]. These data suggested that low expression of miR-135b in the primary tumors may be a risk factor for bone metastatic [71]. 

Among miRNAs identified in this study, many of them were extensively investigated previously, and we confirmed their results. However, the role of some miRNAs in prostate cancer was not previously reported. This may be partly due to differences in methodology, platforms used, or sample size. These miRNAs observed in this study may help to investigate potential mechanisms in different PCa outcomes between AA and EA men. There are some limitations to this study. First, the small number of PCa samples is a limitation. Therefore, these results need to be validated in larger studies. Furthermore, our results were based on analyses of radical prostatectomy specimens; whereas, a future diagnostic test for PCa should use more clinically relevant non-invasive sample types, such as urine or blood.

## 5. Conclusions

In summary, miRNA expression studies provide evidence for the role of miRNAs in PCa diagnosis, prognosis, and elimination of health disparities. Despite these promising studies, there is currently a limited number of PCa-related miRNAs used in the clinical setting. Our findings underscore an important opportunity for the implementation of miRNA-based biomarkers, including miR-1, -30e, -34c, -96, -135b, -139, -145, -183, -299-3p, and -375 for diagnosis, prognosis, and elimination of health disparities for men with PCa. 

## Figures and Tables

**Figure 1 cancers-15-02331-f001:**
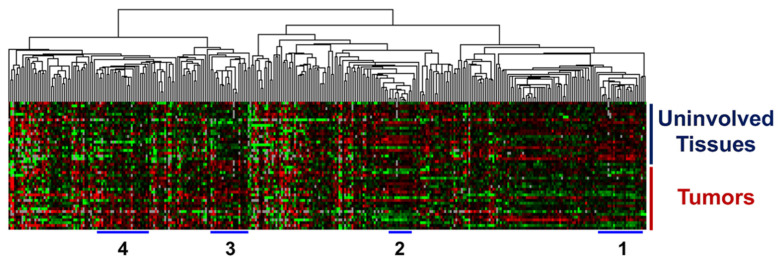
Clustering of miRNA expressions in tumor tissues and uninvolved prostate tissues. Hierarchal clustering of log2-transformed relative expression values of miRNAs in uninvolved prostate and tumor tissues.

**Figure 2 cancers-15-02331-f002:**
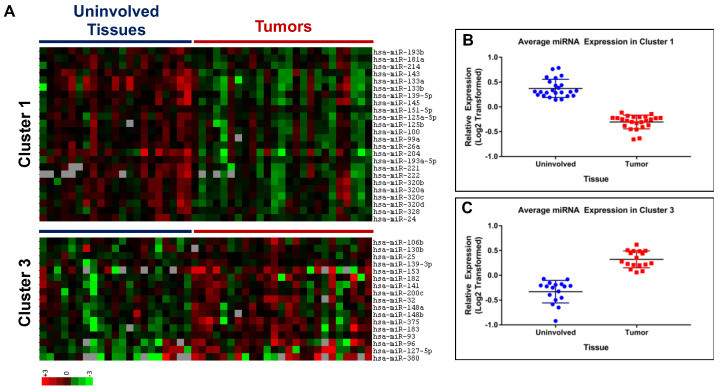
Analysis of clusters 1 and 3 identified from miRNA profiling of prostate tissues. (**A**) Heat maps of expression data for miRNAs in clusters 1 and 3; uninvolved tissue samples grouped under the blue line (left side) and tumor samples grouped under the red line (right side). (**B**,**C**) The average expression value of each miRNA was calculated for uninvolved tissue samples (blue circles) and tumor samples (red squares). These values were used to generate dot plots for cluster 1 (**B**) (*p* = 4.83 × 10^−18^) and cluster 3 (**C**) (*p* = 1.70 × 10^−10^).

**Figure 3 cancers-15-02331-f003:**
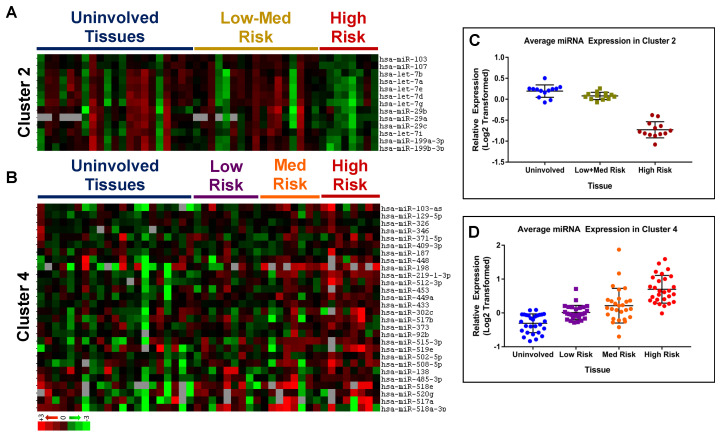
Analysis of clusters 2 and 4 identified from miRNA profiling of prostate tissues. Correlation of miRNA expressions with CAPRA-S risk group. (**A**,**B**) Heat maps of expression data for miRNAs in clusters 2 (**A**) and 4 (**B**). (**A**) Uninvolved tissue samples (U) grouped under the blue line, low- and medium-risk (L+M) patient tumor samples grouped under the gold line, and high-risk (H) patient tumor samples grouped under the red line. (**B**) Uninvolved tissue samples appear under the blue line, low-risk patient tumor samples are grouped under the purple line, medium are grouped under the orange line, and high-risk patient tumor samples are grouped under the red line. (**C**,**D**) The average expression value of each miRNA was calculated for uninvolved tissues and tumor tissues grouped by CAPRA-S risk. Values were used to generate dot plots for cluster 2 (**C**) (*p* = 0.024 U vs. L+M, 2.14 × 10^−12^ U vs. H, 1.57 × 10^−10^ L+M vs. H) and cluster 4 (**D**) (*p* = 1.46 × 10^−5^ U vs. L, 2.60 × 10^−5^ U vs. M, 2.51 × 10^−14^ U vs. H, 0.06 L vs. M, 1.01 × 10^−9^ L vs. H, 0.0002, M vs. H).

**Figure 4 cancers-15-02331-f004:**
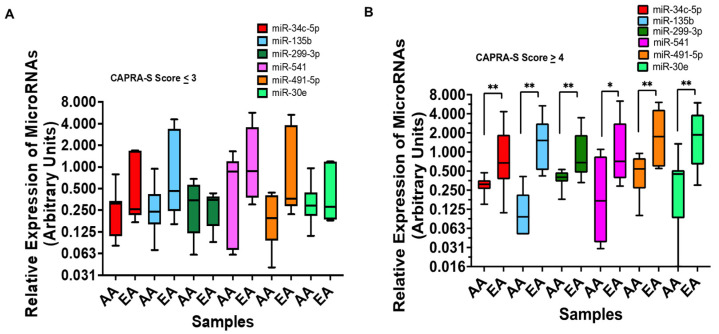
Top miRNAs with differential regulation between European American (EA) and African American (AA) men: (**A**,**B**) the fold change in expression of AR-regulating miRNAs grouped by race. Patients with CAPRA-S scores lower than 3 (**A**) and greater than 3 (**B**) are presented (*p* values: * < 0.07, ** < 0.05.

**Table 1 cancers-15-02331-t001:** Selected characteristics of patients.

Patient ID	Age	Race	PSA	Gleason Score	SM ^1^	ECE ^2^	LN Invasion	SVI ^3^	Stage	CAPRA-S Score	Risk
1	78	White	14.3	3 + 2 = 5	Neg	Neg	Neg	Neg	PT3NOMO	2	low
2	43	Hispanic	5.9	3 + 3 = 6	Pos	Neg	Neg	Neg	PT2CNXMX	2	low
3	53	Black	4.3	3 + 3 = 6	Pos	Neg	Neg	Neg	T2cR1NXMX	2	low
4	69	White	8.2	3 + 4 = 7	Neg	Neg	Neg	Neg	PT2CNOMX	2	low
5	62	White	7.8	3 + 4 = 7	Neg	Neg	Neg	Neg	PT2CNOMX	2	low
6	40	Black	8.8	3 + 4 = 7	Neg	Neg	Neg	Neg	PT2CNXMX	2	low
7	58	White	6.6	3 + 4 = 7	Neg	Neg	Neg	Neg	PT2CNXMX	2	low
8	61	White	3.7	3 + 4 = 7	Neg	Pos	Neg	Neg	PT3ANOMX	2	low
9	69	Black	23.3	3 + 3 = 6	Neg	Neg	Neg	Neg	PT2NOMX	3	med
10	60	NA	6.3	3 + 4 = 7	Pos	Neg	Neg	Neg	PT3BNOMX(IV)	3	med
11	67	White	6.2	3 + 4 = 7	Pos	Neg	Neg	Neg	PT2CR1NXMX	3	med
12	72	Black	4.7	3 + 4 = 7	Neg	Neg	Neg	Pos	T3bN0MX	3	med
13	61	UK ^4^	5.1	3 + 3 = 6	Pos	Neg	Neg	Pos	PT3BNOMX	4	med
14	54	Black	87.4	3 + 3 = 6	Neg	Pos	Neg	Neg	PT3aN0MX	4	med
15	61	Black	9.8	3 + 4 = 7	Pos	Neg	Neg	Neg	PT3AR1NOMX	4	med
16	48	Black	9.4	3 + 4 = 7	Pos	Neg	Neg	Neg	PT2CNOMX	4	med
17	65	Black	8.8	3 + 4 = 7	Pos	Neg	Neg	Neg	PT2cNXMX	4	med
18	61	White	5.4	4 + 3 = 7	Neg	Neg	Neg	Pos	PT3BNOMX	4	med
19	48	Black	6.5	3 + 4 = 7	Pos	Pos	Neg	Neg	T1cNXMX	5	med
20	53	White	8.5	3 + 4 = 7	Pos	Pos	Neg	Neg	PT3aR1NXMX	5	med
21	63	White	4.8	3 + 4 = 7	Pos	Pos	Pos	Pos	PT3BR1N1MX	7	high
22	62	Black	14.9	3 + 4 = 7	Pos	Neg	Neg	Pos	pT3bN0MX	7	high
23	54	White	13.9	4 + 3 = 7	Pos	Pos	Neg	Neg	PT3bR1N0MX	7	high
24	60	White	5.6	4 + 3 = 7	Pos	Pos	Pos	Pos	PT3bN1MX	8	high
25	64	White	51.8	4 + 5 = 9	Pos	Pos	Neg	Neg	NA	9	high

^1.^ SM: surgical margin ^2.^ ECE: extra-capsular extension ^3.^ SVI: seminal vesicle invasion ^4.^ UK: unknown.

## Data Availability

miRNA expression data and clinical data are available upon request.

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
