# Peer review of "Differential Expression of miRNAs Contributes to Tumor Aggressiveness and Racial Disparity in African American Men with Prostate Cancer"

_cancers, 2023, doi:10.3390/cancers15082331_

Round 1

Reviewer 1 Report (Previous Reviewer 1)

The authors have addressed all questions properly and it's ready to publish in cancers. 

This manuscript is a resubmission of an earlier submission. The following is a list of the peer review reports and author responses from that submission.

Round 1

Reviewer 1 Report

The manuscript entitled “Differential expression of miRNAs contributes to tumor aggressiveness and racial disparity in African American men with prostate cancer” by Ottman R et al. studied differential expressed (DE) miRNA identified from prostate cancer patients and looked at the correlation between tumor aggressiveness/racial disparity and expression level of some DE miRNAs. Those DE miRNAs could be applied as potential biomarkers for prostate cancer diagnosis and prognosis. Following questions need to be addressed:

1.       Introduction doesn’t provide sufficient background information such as prostate cancer (patients number, stage, therapeutics), research progress of miRNA on cancer field especially prostate cancer, etc.

2.        The authors utilized tumor tissues and adjacent non-involved areas for miRNA expression analysis in this study. Please define “non-involved areas” with more detail in the method section. Besides, the patient number (n=25) is relative limited to get reliable results.  

3.       Please define all the abbreviations in the manuscript. Some are missing such as PSA, CAPRA-S at line106.

4.       In Figure1, the authors used cluster analysis to identify DE miRNAs. How is it implemented? What’s the criteria to cluster some miRNA together? More detail should be provided to explain the analysis approach.

5.       At line154, it is mentioned that 6 miRNAs are >2-fold down-regulated in malignant tissues. What’s the p-value? Statistical analysis is lacking throughout the whole manuscript. Please provide the p-value for all the plots.

6.       Figure 2A seems wrong-inconsistent with figure legend.

7.       For each >2 fold DE genes, separate expression plot should be provided.

8.       What dose the line mark for each miRNA in Figure 4? Is it AVE of all samples or only samples with CAPRA-S Score <=3? Variation seems very large in European American CAPRA-S Score >=4 groups which makes it hard to get the conclusion.

Reviewer 2 Report

The authors have done a great job on analyzing miRNA from human patients. They have extracted RNA from FFPE tissues and did a cluster analysis for differential miRNA expression. They have discovered interesting and clinically relevant miRNAs for diagnosis or future treatment of high risk prostate cancer patients and how it is different for African American and European men.

1. Line 75 "by a pathologist andtumor". Please proof read.

2. Please make the font size of all the labels of the same size, please increas the size of the smaller font labels in Fig 2, 3 and 4.

3. Please consider transgender women in future studies.